# Prion propagation can occur in a prokaryote and requires the ClpB chaperone

Andy H Yuan[1,2], Sean J Garrity[1], Entela Nako[1], Ann Hochschild[1]*

[1]Department of Microbiology and Immunobiology, Harvard Medical School, Boston, United States; [2]Whitehead Institute for Biomedical Research, Cambridge, United States

**Abstract** Prions are self-propagating protein aggregates that are characteristically transmissible. In mammals, the PrP protein can form a prion that causes the fatal transmissible spongiform encephalopathies. Prions have also been uncovered in fungi, where they act as heritable, protein-based genetic elements. We previously showed that the yeast prion protein Sup35 can access the prion conformation in *Escherichia coli*. Here, we demonstrate that *E. coli* can propagate the Sup35 prion under conditions that do not permit its de novo formation. Furthermore, we show that propagation requires the disaggregase activity of the ClpB chaperone. Prion propagation in yeast requires Hsp104 (a ClpB ortholog), and prior studies have come to conflicting conclusions about ClpB's ability to participate in this process. Our demonstration of ClpB-dependent prion propagation in *E. coli* suggests that the cytoplasmic milieu in general and a molecular machine in particular are poised to support protein-based heredity in the bacterial domain of life.

## Introduction

Prions are infectious, self-propagating protein aggregates first described in the context of scrapie (*Prusiner, 1982*), an example of a class of devastating neurodegenerative diseases known as the transmissible spongiform encephalopathies (TSEs). Specifically, the prion form of a protein known as PrP is the causative agent of the TSEs, which afflicts humans and other mammals. Native PrP (PrP$^C$) undergoes a dramatic change in conformation upon conversion to its prion form (PrP$^{Sc}$), forming distinctive cross-β aggregates termed as amyloid (*Diaz-Espinoza and Soto, 2012*). Highly resistant to denaturation and proteolysis, PrP$^{Sc}$ is infectious and templates the conformational conversion of PrP$^C$ molecules (*Caughey et al., 2009*).

Prion-like phenomena have also been described in budding yeast and other fungi. Since Wickner first invoked prions to account for two examples of non-Mendelian genetic elements in *Saccharomyces cerevisiae* (*Cox, 1965*; *Aigle and Lacroute, 1975*; *Wickner, 1994*), the study of fungal prion proteins has resulted in profound advances in the understanding of prion biology, including the first demonstration that purified prion protein aggregates are infectious (*Maddelein et al., 2002*; *King and Diaz-Avalos, 2004*; *Tanaka et al., 2004*). In general, such prion proteins exist in either a native, soluble form or a self-perpetuating, amyloid form with spontaneous conversion between forms representing a rare event (*Allen et al., 2007*; *Lancaster et al., 2010*). However, unlike PrP$^{Sc}$, yeast prions do not normally cause cell death. Instead, they can act as protein-based genetic elements that confer new phenotypes on those cells that harbor them (*True and Lindquist, 2000*; *Tuite and Serio, 2010*; *Newby and Lindquist, 2013*). Fungal prion proteins have been found to participate in diverse cellular processes (*Coustou et al., 1997*; *True et al., 2004*; *Suzuki et al., 2012*; *Holmes et al., 2013*). The conversion of these proteins to their prion forms typically results in a dominant loss-of-function phenotype (*Cox,*

*For correspondence:
ahochschild@hms.harvard.edu

Competing interests: The authors declare that no competing interests exist.

**eLife digest** Unlike most infectious agents—such as viruses or bacteria—that contain genetic material in the form of DNA or RNA, a prion is simply an aggregate of misfolded proteins. Although they are not living organisms, these prion aggregates can self-propagate; when they enter a healthy organism, they cause existing, correctly folded proteins to adopt the prion fold. Within the aggregate, the prion proteins have a corrugated structure that allows them to stack together tightly, which in turn makes the aggregates very stable. As more prions are formed, they then trigger other protein molecules to misfold and join the aggregates, and the aggregates continue to grow and spread within the infected organism causing tissue damage and cell death.

Prion diseases are well known in mammals, where the prion aggregates typically destroy tissue within the brain or nervous system. Bovine spongiform encephalopathy (also commonly known as BSE or 'mad cow disease') is an example of a prion disease that affects cattle and can be transmitted to humans by eating infected meat. Prions also form in yeast and other fungi. These prions, however, do not cause disease or cell death; instead, yeast prions act as protein-based elements that can be inherited over multiple generations and which provide the yeast with new traits or characteristics. Although prions can form spontaneously in yeast cells, their stable propagation depends on so-called chaperone proteins that help to remodel the prion aggregates. Previous work has shown that bacterial cells can also support the formation of prion-like aggregates. The bacteria were engineered to produce two yeast prion proteins—one of which spontaneously formed aggregates that were needed to trigger the conversion of the other to its prion form. However, it was not known if bacterial cells could support the stable propagation of prions if the initial trigger for prion conversion was removed.

Yuan et al. now reveal that the bacterium *Escherichia coli* can propagate a yeast prion for over a hundred generations, even when the cells can no longer make the protein that serves as the trigger for the initial conversion. This propagation depends on a bacterial chaperone protein called ClpB, which is related to another chaperone protein that is required for stable prion propagation in yeast. As such, the findings of Yuan et al. raise the possibility that, even though a prion specific to bacteria has yet to be identified, prions or prion-like proteins might also contribute to the diversity of traits found in bacteria. Furthermore, since both yeast and bacteria form and propagate prions in similar ways, such protein-based inheritance might have evolved in these organisms' common ancestor over two billion years ago.

---

*1965*; *Aigle and Lacroute, 1975*). A particularly well-characterized example involves the essential translation release factor Sup35, which confers on cells a heritable nonsense suppression phenotype upon conversion to the prion form (*Cox, 1965*; *Ter-Avanesyan et al., 1994*; *Patino et al., 1996*; *Paushkin et al., 1996*).

Like other yeast prion proteins, Sup35 has a modular structure with a distinct prion domain (PrD) that mediates conversion to the prion form, [*PSI*⁺]. In the case of Sup35, the essential prion determinants, which include a glutamine- and asparagine-rich segment and five complete copies of an imperfect oligopeptide repeat sequence, lie in the N-terminal domain (N), whereas translation release activity resides in the C-terminal domain (C) (*Ter-Avanesyan et al., 1993*). A highly charged middle region (M) increases the solubility of native Sup35 and enhances the mitotic stability of [*PSI*⁺] (*Liu et al., 2002*). Together, Sup35 N and M function as a transferable prion-forming module (NM) that maintains its prionogenic potential when fused to heterologous proteins (*Li and Lindquist, 2000*). A distinctive property of the Sup35 conversion process in yeast is its dependence on the presence of a pre-existing prion, designated [*PIN*⁺] for [*PSI*⁺] inducibility factor (*Derkatch et al., 1997*). Thus, yeast strains containing Sup35 in the non-prion form, [*psi*⁻], support the spontaneous conversion to [*PSI*⁺] only if they contain [*PIN*⁺], typically the prion form of the Rnq1 protein (*Derkatch et al., 2000*). However, several other yeast prion proteins, including the New1 protein, have the capacity to function as [*PIN*⁺] in their prion forms (*Derkatch et al., 2001*; *Osherovich and Weissman, 2001*).

Importantly, the stable propagation of yeast prions—and thus, the heritability of their associated phenotypes—depends on the function of chaperone proteins (*Chernoff et al., 1995*). Specifically, the AAA+ disaggregase Hsp104 is strictly required for the propagation of virtually all yeast prions

characterized thus far, and several other chaperone proteins have been implicated in this process as well (*Liebman and Chernoff, 2012*; *Winkler et al., 2012*). Various lines of evidence support the view that the essential role of Hsp104 with respect to prion propagation stems from its ability to fragment prion aggregates and thereby to generate smaller seed particles known as propagons that can be efficiently partitioned to daughter cells during cell division (*Paushkin et al., 1996*; *Ness et al., 2002*; *Cox et al., 2003*; *Kryndushkin et al., 2003*; *Satpute-Krishnan et al., 2007*; *Higurashi et al., 2008*). Accordingly, depletion or inhibition of Hsp104 in a prion-containing cell leads to prion loss in progeny cells.

The molecular processes underlying prion biology constitute at least two distinct phases, namely, (i) the de novo conversion of a protein from its native to prion form, and (ii) the subsequent propagation of the self-perpetuating prion form over multiple generations. While studies have demonstrated that the bacterial cytoplasm can support the de novo formation of prion-like aggregates (*Sabaté et al., 2009*; *Fernándes-Tresguerres et al., 2010*; *Garrity et al., 2010*; *Espargaró et al., 2012*; *Gasset-Rosa et al., 2014*), evidence for prion propagation—and thus, protein conformation-dependent heredity—in bacteria has remained elusive. We previously demonstrated that conversion of Sup35 NM to its prion form in *Escherichia coli*, as in *S. cerevisiae*, depends on [*PIN*⁺], which is formed by providing the bacterial cells with the yeast New1 protein (*Garrity et al., 2010*). This [*PIN*⁺] dependence provides an experimental framework for distinguishing between the initial conversion and subsequent propagation phases of the prion cycle. In particular, the formation of prion-like Sup35 NM aggregates can be induced in bacterial cells containing the New1 protein; subsequent depletion of the New1 protein from these cells reveals whether or not the bacterial cytoplasm can support the propagation of the Sup35 NM prion in the absence of [*PIN*⁺].

Here we show that bacteria can propagate the Sup35 prion in an infectious conformation over at least ~100 generations under conditions that do not permit de novo prion formation. More specifically, we demonstrate maintenance of the Sup35 NM prion over multiple rounds of restreaking in *E. coli* cells no longer capable of synthesizing the New1 protein. Furthermore, we establish that propagation of the Sup35 NM prion in *E. coli* requires the disaggregase activity of ClpB, the bacterial ortholog of Hsp104. The striking parallel between the requirements for both prion formation and prion propagation in yeast and bacteria, which are thought to have diverged more than 2.2 billion years ago, suggests that the paradigm of protein-based heredity may be more ancient than previously inferred (*DeSantis et al., 2012*).

## Results

### *E. coli* cells can propagate SDS-stable Sup35 NM aggregates

Having previously shown that Sup35 NM can adopt an infectious amyloid conformation in the *E. coli* cytoplasm (*Garrity et al., 2010*), we wished to determine whether or not *E. coli* cells could stably propagate Sup35 NM in its prion form. To address this question, we took advantage of the fact that conversion of Sup35 NM to its prion conformation in *E. coli* depends on the presence of New1, mirroring features of the [*PIN*⁺] dependence of Sup35 prion formation in *S. cerevisiae* (*Figure 1A*). Thus, our plan was to induce the formation of infectious Sup35 NM aggregates in *E. coli* cells containing the prionogenic module of New1 and then to monitor the fate of Sup35 NM over multiple generations after curing the cells of New1-encoding DNA.

To facilitate these experiments, we fused Sup35 NM and New1 to two monomeric fluorescent proteins (mCherry bearing a C-terminal hexahistidine tag and mGFP, respectively). The two fusion proteins were produced from compatible plasmids under the control of IPTG-inducible promoters. The plasmid encoding New1-mGFP (pSC101$^{TS}$-*NEW1*) bore a temperature-sensitive origin of replication, enabling us to cure cells of New1-encoding DNA and thereby deplete cells of the New1 fusion protein. As an initial test of our experimental system, we introduced the plasmid encoding Sup35 NM-mCherry-His$_{6x}$ (pBR322-*SUP35 NM*) together with either pSC101$^{TS}$-*NEW1* or an empty vector control (pSC101$^{TS}$) into *E. coli* cells and induced the synthesis of the fusion proteins at the permissive temperature. After overnight growth, we detected SDS-stable Sup35 NM aggregates ('Materials and methods') only in cells producing the New1 fusion protein (*Figure 1B*). As New1 can independently adopt an amyloid conformation in *E. coli* (*Garrity et al., 2010*), we also detected SDS-stable New1 aggregates in cells containing both fusion proteins (*Figure 1B*). Western blot analysis revealed that the intracellular levels of the Sup35 NM fusion protein were comparable in the presence and absence of

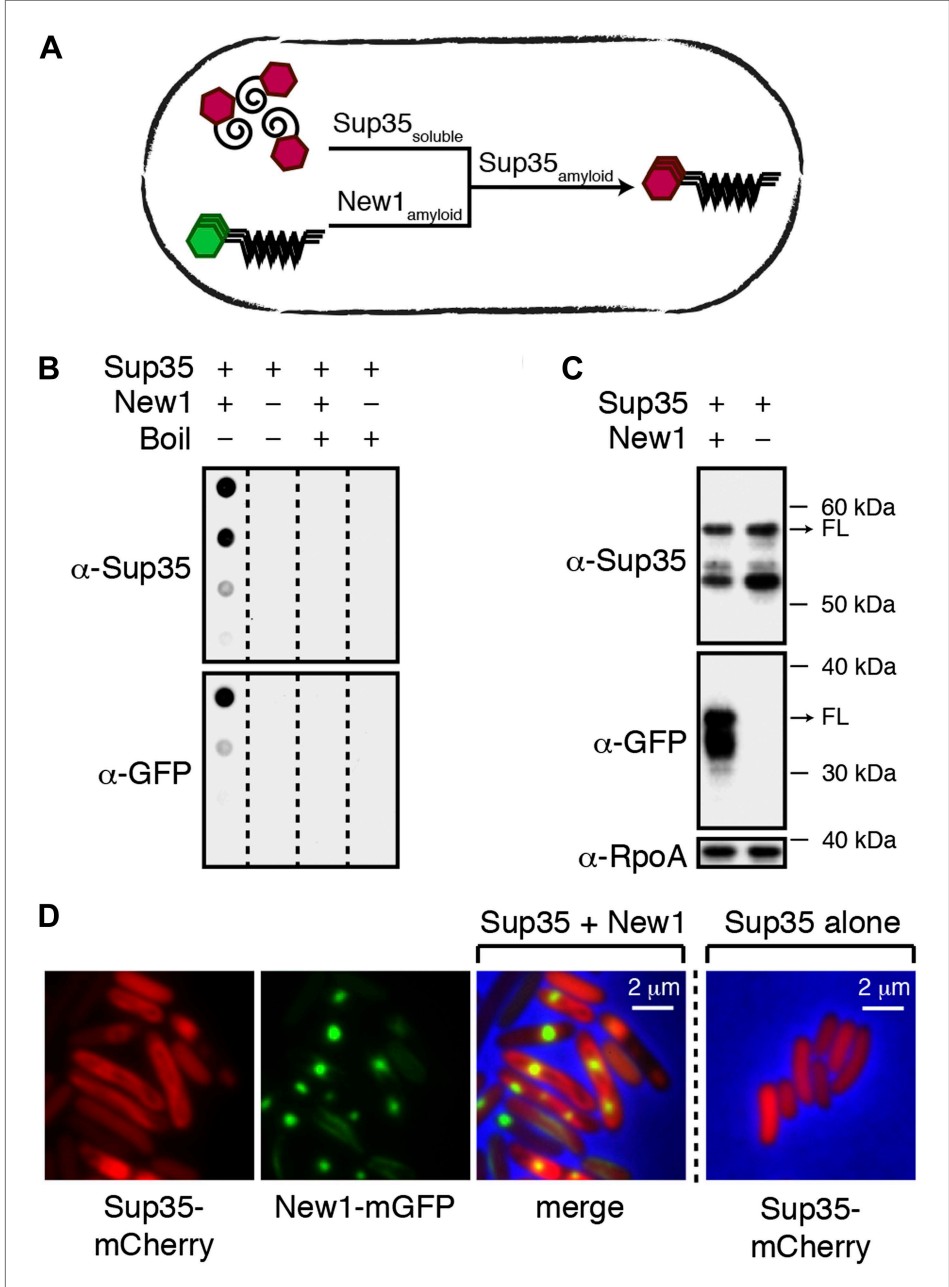

**Figure 1**. Conversion of Sup35 NM to its prion form in *E. coli* requires New1. (**A**) Cartoon representation of how conversion of soluble Sup35 NM (Sup35$_{soluble}$) to its amyloid conformation (Sup35$_{amyloid}$) depends on the presence of New1 in its amyloid conformation (New1$_{amyloid}$). Sup35 NM and New1 (black) are depicted as fusions to mCherry (red) and mGFP (green), respectively. (**B**) SDS-stable Sup35 NM aggregates are detected only in cells producing SDS-stable New1 aggregates as assessed by filter retention analysis. For each sample, undiluted lysate and three twofold dilutions are shown (see 'Materials and methods'). Sup35 NM and New1 aggregates are no longer detected once boiled. The α-Sup35 antibody recognizes the Sup35 NM-mCherry-His$_{6X}$ fusion protein, and the α-GFP antibody detects the New1-mGFP fusion protein. (**C**) Intracellular full-length (FL) Sup35 NM fusion protein levels are comparable in the presence and absence of New1 as assessed by Western blot analysis. The α-RpoA antibody recognizes the α subunit of *E. coli* RNA polymerase. (**D**) Fluorescence images of representative cells containing Sup35 NM and New1 or Sup35 NM alone. For cells containing both fusion proteins, the mCherry channel, GFP channel, and merged images are shown.

The following figure supplement is available for figure 1:

*Figure 1. Continued on next page*

*Figure 1. Continued*

**Figure supplement 1**. α-Sup35 and α-His$_{6X}$ antibodies are interchangeable for detecting the Sup35 NM-mCherry-His$_{6X}$ fusion protein.

the New1 fusion protein (*Figure 1C*, *Figure 1—figure supplement 1A*). We also examined cells by fluorescence microscopy. In cells containing both fusion proteins, Sup35 NM formed twisted ring structures (*Garrity et al., 2010*) or large polar foci in 22.5% of cells (*N* = 258), whereas New1 formed punctate foci in 89.8% of cells (*N* = 258). (*Figure 1D*). In contrast, Sup35 NM exhibited diffuse fluorescence in 100% of cells lacking New1 (*N* = 532) (*Figure 1D*).

We then sought to determine whether or not *E. coli* cells could propagate SDS-stable Sup35 NM aggregates over multiple generations under conditions that do not permit the de novo formation of aggregates (that is, in the absence of New1). Our experimental protocol is illustrated in *Figure 2* (see also *Figure 3A*). We first induced fusion protein synthesis in cells transformed with pBR322-*SUP35 NM* and either pSC101$^{TS}$-*NEW1* (experimental sample) or pSC101$^{TS}$ (control sample). These 'starter cultures' were grown overnight to allow for the formation of SDS-stable Sup35 NM aggregates in the experimental sample. The cells were then plated and grown at the non-permissive temperature to cure the cells of pSC101$^{TS}$-*NEW1* or pSC101$^{TS}$, thereby generating a set of Round 1 (R1) colonies. 20 R1 experimental colonies and 20 R1 control colonies were subsequently examined; each was (a) patched onto selective medium to test for loss of pSC101$^{TS}$-*NEW1* or pSC101$^{TS}$, (b) restreaked to generate Round 2 (R2) colonies, and (c) inoculated into liquid medium for overnight growth to test for the presence of SDS-stable Sup35 NM aggregates. Four separate experimental lineages (L1$_E$–L4$_E$) originating from ancestral R1 experimental colonies containing detectable Sup35 NM aggregates along with four separate control lineages (L1$_C$–L4$_C$) originating from ancestral R1 control colonies were then followed through Round 3 (R3) and Round 4 (R4). For R2 and each subsequent round, 10 experimental colonies and 10 control colonies were analyzed.

All R1 experimental and control colonies (20 of each) had lost pSC101$^{TS}$-*NEW1* or pSC101$^{TS}$, respectively, as assessed by patching on selective medium (data not shown). Moreover, the absence of *NEW1* DNA was confirmed by PCR (*Figure 3C*) and the absence of New1 protein was confirmed by Western blot analysis (*Figure 3B*). We detected SDS-stable Sup35 NM aggregates in 8 of 20 experimental samples (*Figure 3B*) and none of the control samples (*Figure 3D*). We selected 4 of the 8 aggregate-positive clones (*Figure 3B*, asterisks) to establish the four experimental lineages and arbitrarily selected four aggregate-negative control clones (*Figure 3D*, asterisks) to establish the four control lineages. 2 of the 4 experimental lineages (L1$_E$ and L3$_E$) retained SDS-stable Sup35 NM aggregates throughout the course of the experiment (*Figure 4A*, *Figure 4—figure supplement 1B*). Of these two lineages, one maintained aggregates in 9 of 10 R4 clones (*Figure 4A*) and the other maintained aggregates in 7 of 10 R4 clones (*Figure 4—figure supplement 1B*). We conclude that SDS-stable Sup35 NM aggregates can be propagated in *E. coli* for at least ~100 generations in the absence of New1 (*Figure 5A*).

The remaining two experimental lineages (L2$_E$ and L4$_E$) lost detectable SDS-stable Sup35 NM aggregates at R3 and R4, respectively (*Figure 4—figure supplement 1A,C*, *Figure 5A*). Moreover, the loss of aggregates manifested itself in all ten of the selected colonies at either R3 or R4 (*Figure 4—figure supplement 1A,C*, *Figure 5A*). Curiously, the loss of SDS-stable Sup35 NM aggregates from a particular lineage coincided with a loss of detectable fusion protein, as assessed by Western blot analysis (*Figure 4—figure supplement 1A,C*), suggesting that the loss of aggregates represented an indirect consequence of a radical drop in protein levels (see 'Discussion'). We note that the observed drop in Sup35 NM-mCherry-His$_{6x}$ fusion protein levels is not irreversible, as exemplified by L2$_E$, which exhibited a loss of detectable Sup35 NM aggregates coincident with an R3 drop in fusion protein levels and an apparent restoration of fusion protein levels by R4 in all 10 samples (*Figure 4—figure supplement 1A*). Despite the presence of normal levels of fusion protein, SDS-stable Sup35 NM aggregates were not recovered in R4 of L2$_E$, consistent with the expectation that prion propagation requires that prion protein synthesis be maintained above some threshold level (*Figure 4—figure supplement 1A*, *Figure 5A*; *Holmes et al., 2013*). Critically, none of the samples (120 in total) from any of the four control lineages contained detectable SDS-stable Sup35 NM aggregates (*Figure 4B*, *Figure 4—figure supplement 2*).

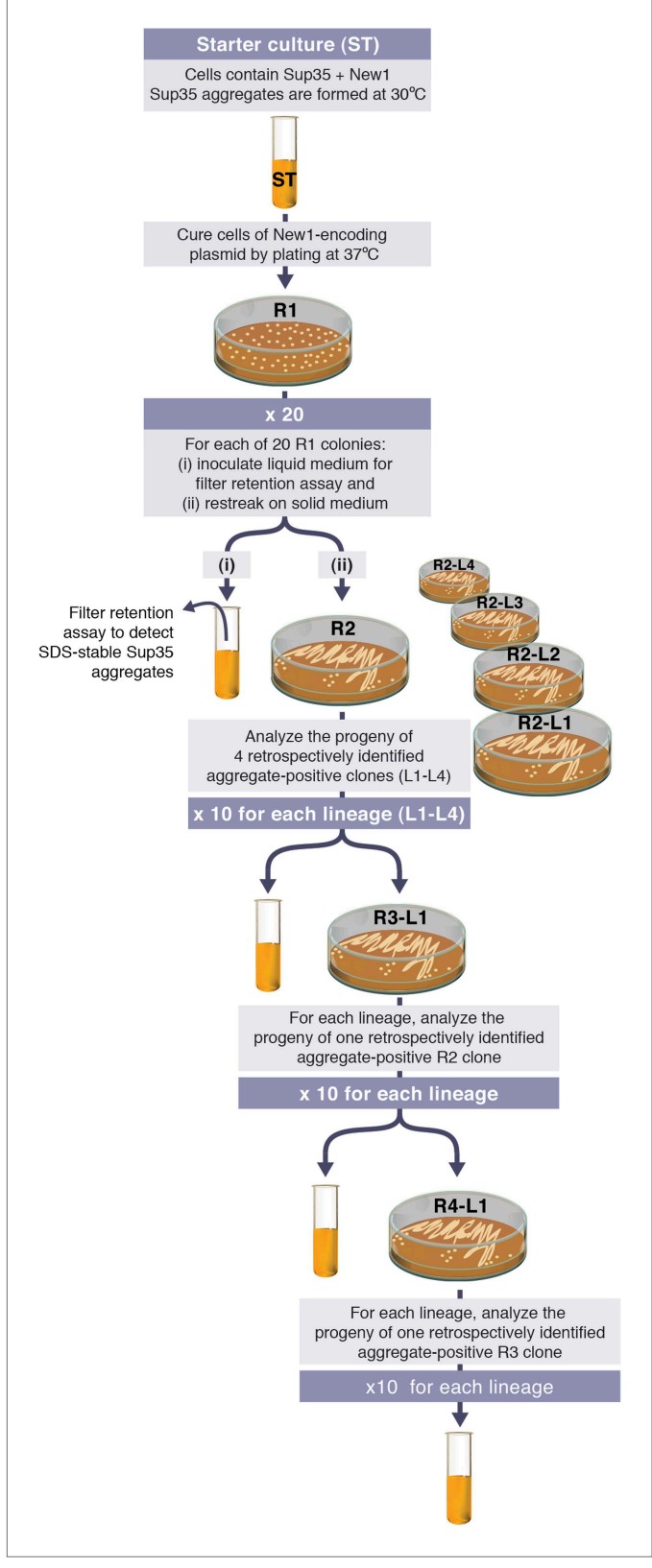

**Figure 2**. Experimental protocol for assessing the ability of *E. coli* cells to propagate SDS-stable Sup35 NM aggregates. Experiments are initiated with either a starter culture (ST) of cells containing Sup35 NM and New1 (shown) or a starter culture of cells containing Sup35 NM alone (not shown). For each of the 4 lineages (L1–L4), the
*Figure 2. Continued on next page*

*Figure 2. Continued*

total number of generations over which Sup35 NM prion propagation is monitored corresponds to the number of cell divisions that occur in the absence of New1 during 4 rounds (R1–R4) of growth on solid medium and an additional round of growth in liquid medium. Growth in the absence of New1 begins at the time the starter culture cells are plated at 37°C (R1). Single R1 colonies were found to contain ~950,000 colony forming units (CFUs), and the liquid cultures contain ~$10^8$ CFUs per μl. Thus, prion propagation is monitored over 98.7 or ~100 generations. We note that the presence of SDS-stable Sup35 NM aggregates in experimental starter culture cells does not represent prion propagation because the presence of New1 (i.e., [*PIN*⁺]) enables the continuing de novo conversion of newly synthesized Sup35 NM to the prion form.

Fluorescence microscopy revealed that cells containing propagated Sup35 NM aggregates exhibited smaller foci emanating from large aggregates typically localized at cell poles, a phenotype distinguished from experimental starter culture cells by the lack of twisted ring structures (*Figure 5C*). However, we observed one instance of aggregate-positive R1 cells exhibiting twisted ring structures (see *Figure 6—figure supplement 2C*). Whereas we cannot definitively assign the SDS-stable Sup35 NM aggregates detected by filter retention to those structures detected by fluorescence microscopy, we note that fluorescence microscopy of prion-containing yeast cells has also revealed structural diversity (*Derkatch et al., 2001*; *Zhou et al., 2001*). Furthermore, cells from aggregate-negative samples invariably exhibited diffuse fluorescence (*Figure 5D*).

## Propagated SDS-stable Sup35 NM aggregates are infectious when introduced into [*psi*⁻] yeast cells

We next sought to determine whether cells that maintained SDS-stable Sup35 NM aggregates in the absence of New1 contained infectious material capable of converting [*psi*⁻] yeast cells to [*PSI*⁺]. We prepared bacterial cell extracts from both experimental and control starter cultures, as well as aggregate-positive samples from each of the four experimental lineages. For each experimental lineage, we examined an arbitrarily chosen sample obtained from the last aggregate-positive round. In addition, for L2$_E$ and L4$_E$, we examined, respectively, the R2 and R3 samples (indicated by asterisks in *Figure 5E*) that gave rise to aggregate-negative clones in the subsequent round. We used these bacterial extracts to transform *S. cerevisiae* spheroplasts prepared from a [*pin*⁻][*psi*⁻] strain. The use of a [*pin*⁻] recipient strain was critical as transient overproduction of Sup35 (or Sup35 NM) in [*PIN*⁺][*psi*⁻] strains significantly stimulates the conversion from [*psi*⁻] to [*PSI*⁺] (*Derkatch et al., 1997*), whereas conversion in a [*pin*⁻][*psi*⁻] background requires the introduction of infectious seed material (*Tanaka and Weissman, 2006*).

The experimental starter culture yielded [*PSI*⁺] yeast transformants at a frequency of ~1% (*Figure 5E*), consistent with our previous findings (*Garrity et al., 2010*). Similarly, each of the aggregate-positive samples from the four experimental lineages yielded [*PSI*⁺] yeast transformants at a frequency of ~1%; in contrast, the aggregate-negative samples yielded no [*PSI*⁺] yeast transformants (*Figure 5E*). Among the [*PSI*⁺] transformants we obtained with the experimental samples, we observed both 'strong' and 'weak' strains (*Figure 5—figure supplement 1*; *Tanaka et al., 2004*; *Frederick et al., 2014*). We note that ~1% corresponds only to the frequency of strong [*PSI*⁺] transformants and therefore represents a conservative estimate of *E. coli* cell extract infectivity; we did not attempt to quantify weak [*PSI*⁺] transformants because they are difficult to distinguish from [*psi*⁻] transformants on the medium utilized to isolate transformants. We conclude that *E. coli* cells can propagate Sup35 NM in an infectious prion conformation over at least ~100 generations under conditions that do not permit de novo prion formation.

## Propagation of infectious Sup35 NM aggregates depends on ClpB

The propagation of [*PSI*⁺] and other prions in yeast requires Hsp104, an Hsp100-family ATP-dependent disaggregase that functions as a ring-shaped hexamer. Specifically, Hsp104 is thought to facilitate prion propagation by fragmenting large aggregates into smaller propagons that are subsequently disseminated during cell division (*Paushkin et al., 1996*; *Ness et al., 2002*; *Cox et al., 2003*; *Kryndushkin et al., 2003*; *Satpute-Krishnan et al., 2007*; *Higurashi et al., 2008*). We therefore investigated whether or not the propagation of Sup35 NM aggregates in *E. coli* requires ClpB, the bacterial ortholog of Hsp104. To address this question, we sought to deplete cells of ClpB specifically during the propagation phase of our experiments. To accomplish this, we modified pSC101$^{TS}$-*NEW1* such that

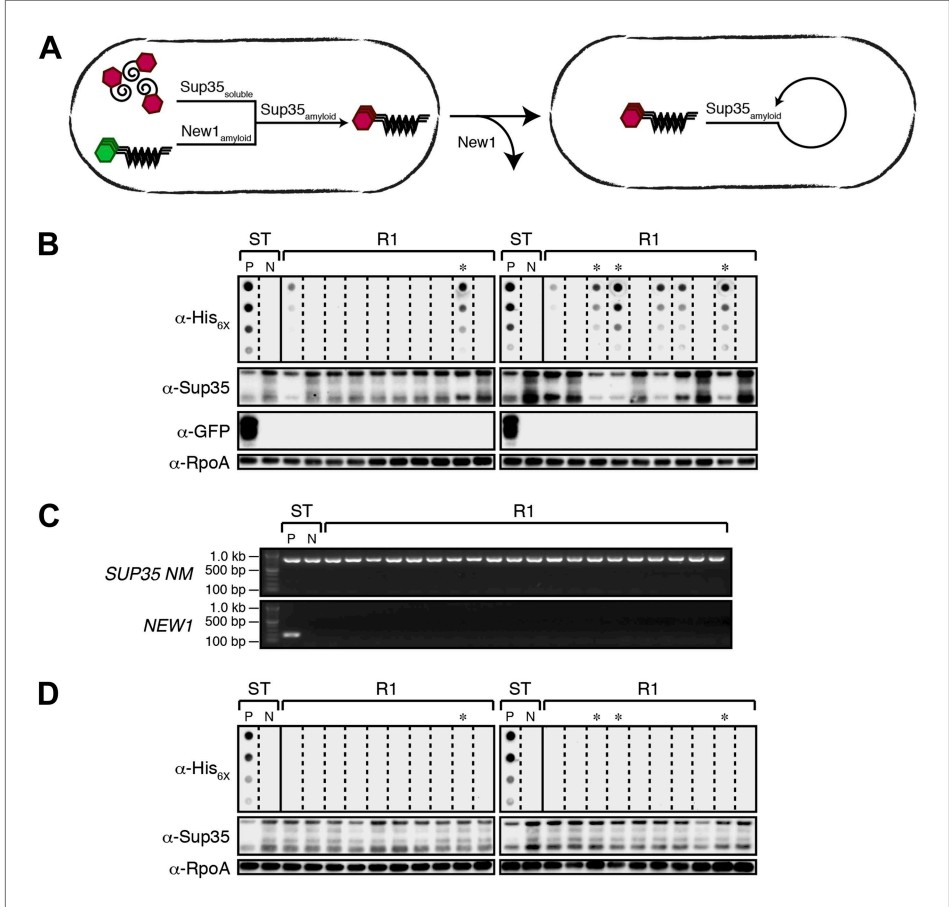

**Figure 3**. Converted Sup35 NM can remain in its prion conformation in *E. coli* cells lacking New1. (**A**) Cartoon representation of how Sup35 NM can convert to its prion form in the presence of New1 and remain in the prion conformation after cells have been cured of New1-encoding DNA. Sup35 NM and New1 (black) are depicted as fusions to mCherry (red) and mGFP (green), respectively. (**B**) SDS-stable Sup35 NM aggregates are detected in 8 of 20 Round 1 (R1) experimental clones derived from a starter culture (ST) of cells containing Sup35 NM and New1 as assessed by filter retention analysis. Starter cultures of cells containing Sup35 NM and New1 and cells containing Sup35 NM alone serve as positive (P) and negative (N) controls, respectively. The four aggregate-positive clones selected to establish the four experimental lineages are indicated by asterisks. In all 20 R1 experimental samples, intracellular Sup35 NM fusion protein levels are comparable, and New1 fusion protein is not detectable as assessed by Western blot analysis. The α-His$_{6X}$ and α-Sup35 antibodies recognize the Sup35 NM-mCherry-His$_{6X}$ fusion protein (see *Figure 1—figure supplement 1B,C*), the α-GFP antibody detects the New1-mGFP fusion protein, and the α-RpoA antibody recognizes the α subunit of *E. coli* RNA polymerase. (**C**) In all 20 R1 experimental samples, DNA encoding Sup35 NM is detectable whereas DNA encoding the prionogenic module of New1 is not detectable by PCR. (**D**) SDS-stable Sup35 NM aggregates are not detected in any of the 20 R1 control clones derived from a starter culture of cells containing Sup35 NM alone as assessed by filter retention analysis. The four aggregate-negative clones selected to establish the four control lineages are indicated by asterisks. Intracellular Sup35 NM fusion protein levels are comparable in all 20 R1 control samples.

it also directed the expression of *clpB* under the control of its native promoter. When transformed into Δ*clpB* cells, pSC101$^{TS}$-*NEW1-clpB* enabled us to grow starter cultures containing ClpB and subsequently to deplete both ClpB and New1 in cells plated at the non-permissive temperature (*Figure 6A*).

As expected, we detected SDS-stable Sup35 NM aggregates in Δ*clpB* starter culture cells transformed with pBR322-*SUP35 NM* and pSC101$^{TS}$-*NEW1-clpB* (*Figure 6—figure supplement 1A*). Furthermore, fluorescence microscopy revealed that these cells contained visible aggregates that were nearly indistinguishable from those in wild-type cells containing pBR322-*SUP35 NM* and pSC101$^{TS}$-*NEW1* (*Figure 6—figure supplement 1B*). After plating Δ*clpB* starter culture cells containing pBR322-*SUP35*

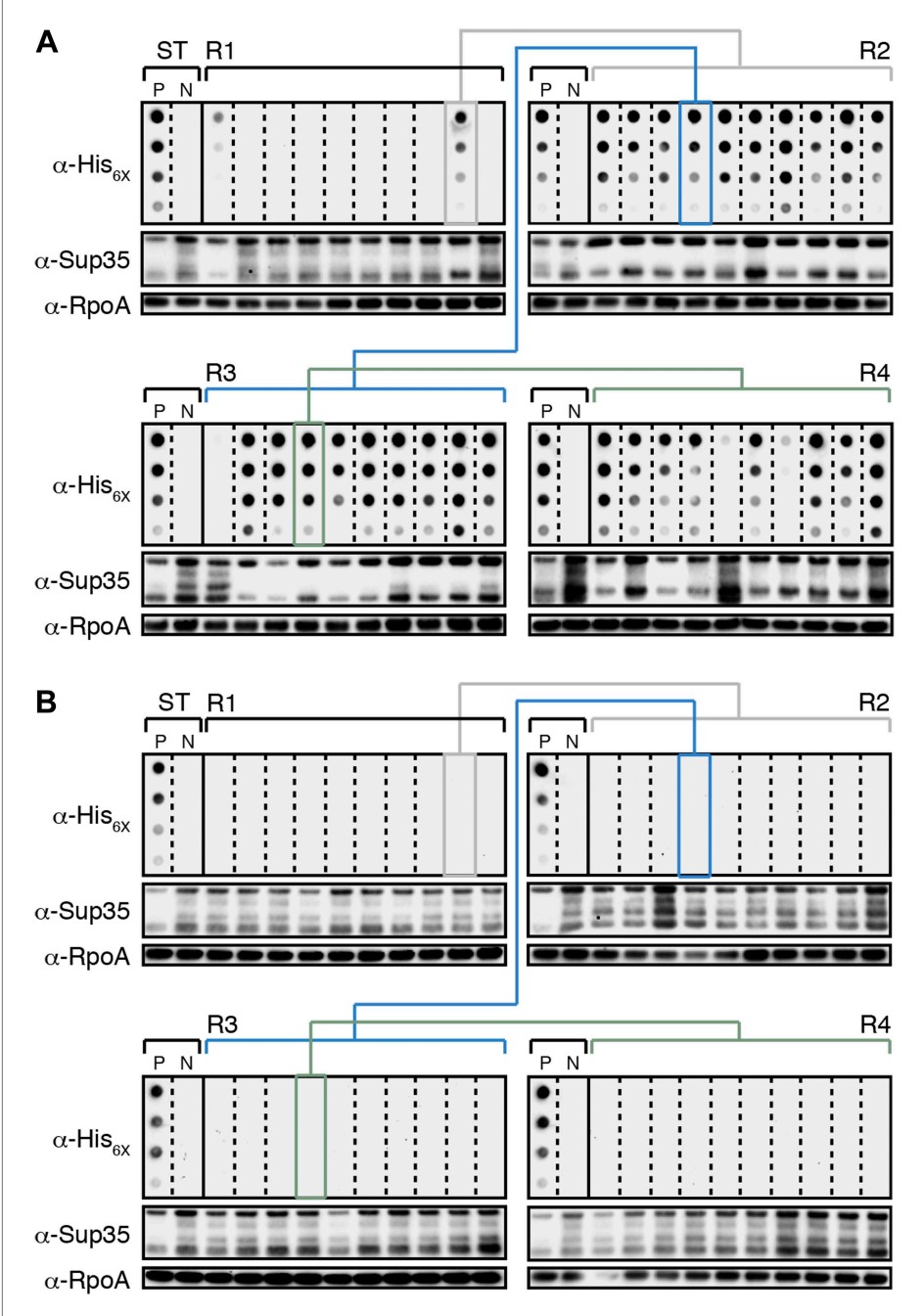

**Figure 4**. *E. coli* can propagate the Sup35 NM prion over ~100 generations. (**A**) Experimental Lineage 1 (L1$_E$). An aggregate-positive Round 1 (R1) experimental clone (gray box) derived from a starter culture (ST) of cells containing Sup35 NM and New1 is identified and restreaked to yield progeny Round 2 (R2) clones (gray bracket). All 10 R2 clones analyzed contain detectable SDS-stable Sup35 NM aggregates. An aggregate-positive R2 clone (blue box) is identified and restreaked to yield progeny Round 3 (R3) clones (blue bracket). Again, all 10 R3 clones analyzed contain SDS-stable Sup35 NM aggregates. An aggregate-positive R3 clone (green box) is identified and restreaked to yield progeny Round 4 (R4) clones (green bracket). 9 of 10 R4 clones analyzed contain SDS-stable Sup35 NM aggregates. The filter retention assay is used to detect SDS-stable Sup35 NM aggregates. Intracellular Sup35 NM fusion protein levels are comparable in all 40 samples as assessed by Western blot analysis. Starter cultures of cells containing Sup35 NM and New1 and cells containing Sup35 NM alone serve as positive (P) and negative (N) controls, respectively. The α-His$_{6X}$ and α-Sup35 antibodies recognize the Sup35 NM-mCherry-His$_{6X}$ fusion protein, and the α-RpoA antibody recognizes the α subunit of *E. coli* RNA polymerase. (**B**) Control Lineage 1 (L1$_C$). An

*Figure 4. Continued on next page*

*Figure 4. Continued*

aggregate-negative R1 control clone (gray box) derived from a starter culture of cells containing Sup35 NM alone is identified and restreaked to yield progeny R2 clones (gray bracket), an aggregate-negative R2 clone (blue box) is identified and restreaked to yield progeny R3 clones (blue bracket), and an aggregate-negative R3 clone (green box) is identified and restreaked to yield progeny R4 clones (green bracket). No SDS-stable Sup35 NM aggregates are detectable in any sample. Intracellular Sup35 NM fusion protein levels are comparable in all 40 samples as assessed by Western blot analysis.

The following figure supplements are available for figure 4:

**Figure supplement 1**. The fate of Sup35 NM in experimental Lineages 2-4.

**Figure supplement 2**. The fate of Sup35 NM in control Lineages 2–4.

---

*NM* and pSC101$^{TS}$-*NEW1-clpB* at the non-permissive temperature to cure cells of ClpB- and New1-encoding DNA, we examined 60 R1 colonies for the presence of SDS-stable Sup35 NM aggregates. In parallel, we examined 60 R1 colonies derived from wild-type starter culture cells containing pBR322-*SUP35 NM* and pSC101$^{TS}$-*NEW1*. As before, every selected colony was patched onto selective medium to test for the loss of pSC101$^{TS}$-*NEW1-clpB* or pSC101$^{TS}$-*NEW1* and inoculated into liquid medium for overnight growth to test for the presence of SDS-stable Sup35 NM aggregates. All selected colonies had lost the appropriate temperature-sensitive vector and the absence of New1 and/or ClpB was confirmed by Western blot analysis (*Figure 6B*, *Figure 6C*). Whereas 17 of 60 (28%) wild-type R1 samples tested aggregate-positive, all Δ*clpB* R1 samples tested aggregate-negative (*Figure 6B*, *Figure 6—figure supplement 2A*, *Figure 6C*, *Figure 6—figure supplement 2B*). Western blot analysis revealed that the wild-type and Δ*clpB* R1 cells contained comparable amounts of Sup35 NM fusion protein (*Figure 6B*, *Figure 6C*). Furthermore, yeast transformation assays confirmed the presence of infectious material capable of converting [*psi*$^-$] yeast cells to [*PSI*$^+$] in Δ*clpB* starter culture cells transformed with pBR322-*SUP35 NM* and pSC101$^{TS}$-*NEW1-clpB* as well as in an aggregate-positive R1 clone derived from wild-type starter culture cells (*Figure 6D*). In contrast, a Δ*clpB* R1 clone derived from Δ*clpB* starter culture cells containing pBR322-*SUP35 NM* and pSC101$^{TS}$-*NEW1-clpB* as well as an aggregate-negative R1 clone derived from wild-type starter culture cells containing pBR322-*SUP35 NM* and pSC101$^{TS}$ lacked detectable infectivity (*Figure 6D*). We conclude that cells lacking ClpB cannot propagate Sup35 NM in its infectious prion conformation.

## ClpB disaggregase activity is required for propagation of SDS-stable Sup35 NM aggregates

To investigate the mechanistic basis for the ClpB dependence of Sup35 NM prion propagation in *E. coli*, we devised a strategy that enabled us to test the abilities of specific ClpB mutants to support the propagation of SDS-stable Sup35 NM aggregates after their formation in the presence of wild-type ClpB. The disaggregase function of ClpB, which assembles as a two-tiered hexameric ring (*Lee et al., 2003*), depends on its abilities to hydrolyze ATP, to translocate polypeptides through its central pore and to collaborate with DnaK (the bacterial Hsp70) and its co-chaperones DnaJ and the nucleotide exchange factor GrpE (reviewed in *Doyle and Wickner, 2009*). Accordingly, we tested previously characterized ClpB mutants specifically defective for (i) ATP hydrolysis (E279A/E678A) (*Weibezahn et al., 2003*) (ii) substrate threading through the ClpB pore (Y653A) (*Weibezahn et al., 2004*), and (iii) collaboration with DnaK (E432A) (*Oguchi et al., 2012*; *Seyffer et al., 2012*; *Carroni et al., 2014*). We note that each of these ClpB mutants is fully proficient for oligomerization and only ClpB E279A/E678A is deficient in ATPase activity (*Mogk et al., 2003*; *Weibezahn et al., 2004*; *Oguchi et al., 2012*). Our strategy required us to construct strains in which we could induce the production of a ClpB mutant specifically during the propagation phase of the experiment while providing wild-type ClpB during the formation phase of the experiment only. To accomplish this, we placed each of the mutant *clpB* alleles (or the wild-type allele) under the control of the anhydrotetracycline (aTc)-inducible promoter P$_{LtetO-I}$ (*Lutz and Bujard, 1997*), integrated these constructs onto the chromosome of our Δ*clpB* strain, and transformed the resulting strains with pBR322-*SUP35 NM* and pSC101$^{TS}$-*NEW1-clpB*.

As expected, we detected SDS-stable Sup35 NM aggregates in starter culture cells of all strains producing plasmid-encoded Sup35 NM-mCherry-His$_{6X}$, New1-mGFP, and wild-type ClpB (*Figure 7A*).

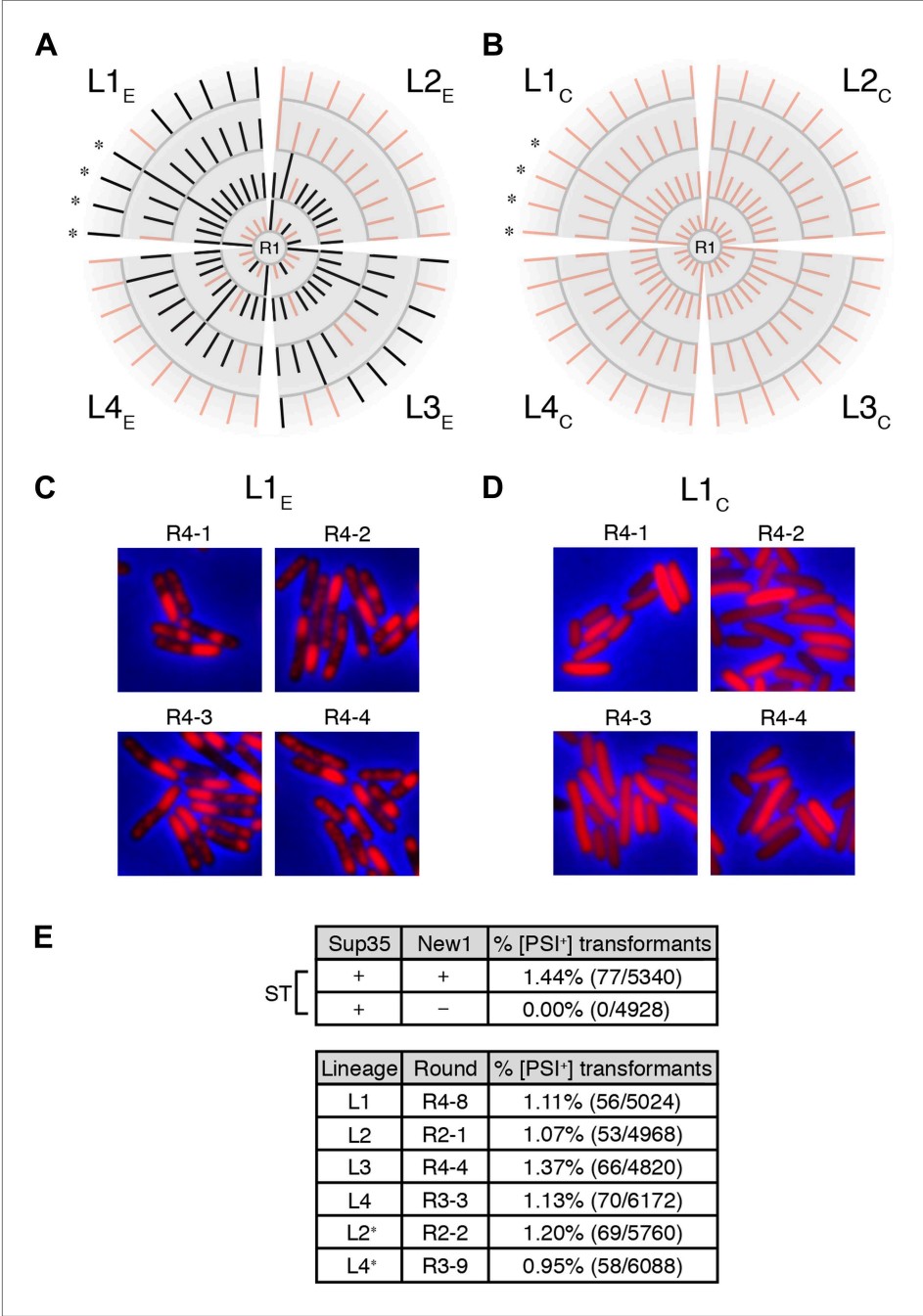

Figure 5. Genealogy of *E. coli* cell lineages propagating Sup35 NM in an infectious prion conformation. (**A**) The fate of Sup35 NM in four experimental lineages (L1$_E$-L4$_E$) established from a starter culture of cells containing Sup35 NM and New1 is shown. Clones that maintain or lose the Sup35 NM prion are indicated by black or pink lines, respectively. Rounds 1–4 (R1–R4) are depicted as gray arcs, with R1 situated at the center of the tree. Clones are designated as aggregate-positive if they contain SDS-stable Sup35 NM aggregates that are detectable in the undiluted sample and at least 1 of the 3 two-fold serial dilutions, as analyzed by filter retention. L1$_E$ and L3$_E$ retain SDS-stable Sup35 NM aggregates for the duration of the experiment (**Figure 4A**, **Figure 4—figure supplement 1B**). L2$_E$ and L4$_E$ lose detectable SDS-stable Sup35 NM aggregates at R3 and R4, respectively. In both cases, the loss of SDS-stable aggregates coincides with a dramatic yet apparently reversible drop in fusion protein levels (**Figure 4—figure supplement 1A,C**; see 'Discussion'). Cells from four aggregate-positive L1$_E$-R4 clones visualized by fluorescence microscopy are indicated by asterisks. (**B**) The fate of Sup35 NM in four control lineages (L1$_C$-L4$_C$) established from a starter culture of cells containing Sup35 NM alone is shown. None of the 120 clones analyzed

*Figure 5. Continued on next page*

*Figure 5. Continued*

contain SDS-stable Sup35 NM aggregates (*Figure 4B*, *Figure 4—figure supplement 2*). Cells from four aggregate-negative L1$_C$-R4 clones visualized by fluorescence microscopy are indicated by asterisks. (**C**) Fluorescence images of representative cells corresponding to the four aggregate-positive R4 clones indicated by asterisks in (**A**). (**D**) Fluorescence images of representative cells corresponding to the four aggregate-negative R4 clones indicated by asterisks in (**B**). (**E**) *E. coli* cell extracts containing propagated, SDS-stable Sup35 NM aggregates are infectious when transformed into *S. cerevisiae* [*psi⁻*] cells. A starter culture (ST) of cells containing Sup35 NM and New1 contain infectious SDS-stable Sup35 NM aggregates capable of converting [*psi⁻*] yeast cells to [*PSI⁺*]. In contrast, a starter culture of cells containing Sup35 NM alone lacks detectable infectivity. Progeny cell extracts transformed into yeast are identified as R*X-Y*, where *X* corresponds to a round number and *Y* corresponds to a clone number assigned sequentially and clockwise according to (**A**) and (**B**). Clones that gave rise to aggregate-negative progeny in the subsequent round are indicated by asterisks. Analysis of these data by Fisher's exact test indicates that the differences in the frequency of [*PSI⁺*] transformants observed with samples containing SDS-stable Sup35 NM aggregates compared with the sample containing soluble Sup35 NM are statistically significant (p < 0.0001). The percentages given refer to strong [*PSI⁺*] transformants; samples containing SDS-stable Sup35 NM aggregates (but not samples containing soluble Sup35 NM) also gave rise to weak [*PSI⁺*] transformants (*Figure 5—figure supplement 1*), but these were not quantified ('Results').

The following figure supplement is available for figure 5:

**Figure supplement 1**. Bacterial cell extracts containing propagated, infectious Sup35 NM aggregates yield both strong and weak [*PSI⁺*] yeast transformants.

---

To determine whether or not each of the mutants could support the propagation of these aggregates following the depletion of New1 and wild-type ClpB, we plated the starter culture cells at the nonpermissive temperature on solid medium lacking or containing increasing concentrations of aTc, generating sets of R1 colonies. We prepared cell extracts from scraped R1 colonies ('Materials and methods') and examined these extracts for the presence or absence of SDS-stable Sup35 NM aggregates. Whereas SDS-stable Sup35 NM aggregates were detected as a function of increasing aTc concentration in cells carrying the wild-type *clpB* allele, no aggregates were detected in cells harboring the *clpB* E279A/E678A, *clpB* Y653A, or *clpB* E432A allele at any concentration of aTc (*Figure 7A*). Western blot analysis revealed that levels of chromosomally-encoded wild-type ClpB and each of the three disaggregase mutants were comparable in cell extracts prepared from colonies scraped off of plates containing 50 ng/ml aTc (*Figure 7B*). Furthermore, replica plating confirmed that all colonies of R1 cells grown on medium supplemented with 50 ng/ml aTc had been cured of pSC101$^{TS}$-*NEW1-clpB* (*Figure 7—figure supplement 1*). We conclude that ATP hydrolysis coupled to substrate translocation through the ClpB central pore and collaboration with DnaK are required for propagation of SDS-stable Sup35 NM aggregates in the absence of New1.

## Discussion

Our findings establish that bacteria can propagate a prion. This is, to our knowledge, the first formal demonstration of prion propagation in a non-eukaryote. More specifically, the [*PIN⁺*]-dependent de novo conversion of Sup35 NM to its prion form in our *E. coli* system enabled us to distinguish experimentally between the initial formation phase and subsequent propagation phase of the prion cycle. Under our experimental conditions, two of four cell lineages (L1$_E$ and L3$_E$) maintained the prion for the duration of the experiment (that is, ~100 generations), one lineage (L4$_E$) maintained the prion for ~80 generations, and one lineage (L2$_E$) maintained the prion for ~60 generations. Furthermore, our work demonstrates that prion propagation in *E. coli* requires the disaggregase activity of ClpB, the bacterial ortholog of Hsp104. We conclude that bacteria can support a chaperone-dependent, protein-based mode of heredity and speculate that the emergence of prion-like phenomena may have predated the evolutionary split between eukaryotes and bacteria.

### Stability of prion propagation in *E. coli*

As only two of four lineages retained the prion for the full duration of our experiments, propagation of the Sup35 NM prion may be less stable in *E. coli* than in *S. cerevisiae* (*DiSalvo et al., 2011*). However, as noted above, loss of the prion in two lineages was coincident with a dramatic drop in Sup35 NM fusion protein levels. Furthermore, this drop was evidently reversible as Sup35 NM fusion protein levels were restored in R4 of L2$_E$ without reappearance of the prion. We do not understand the

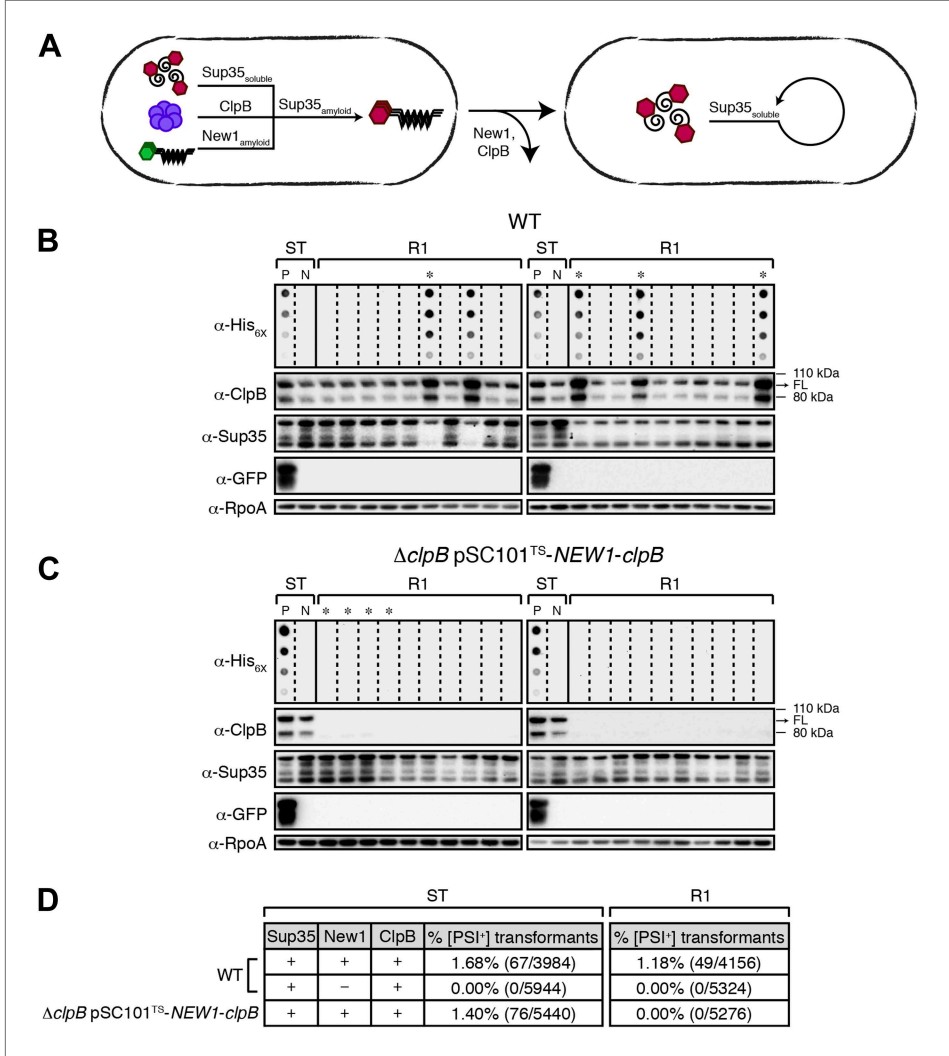

**Figure 6**. Sup35 NM prion propagation in *E. coli* requires ClpB. (**A**) Cartoon representation of how Sup35 NM can convert to its prion form in the presence of New1 and ClpB but cannot propagate in the prion conformation after cells have been cured of New1- and ClpB-encoding DNA. Sup35 NM and New1 (black) are depicted as fusions to mCherry (red) and mGFP (green), respectively. ClpB is depicted as a purple hexamer. (**B**) SDS-stable Sup35 NM aggregates are detected in 5 of 20 Round 1 (R1) wild-type (WT) clones derived from a starter culture (ST) of wild-type cells containing Sup35 NM and New1 as assessed by filter retention analysis. In total, 17 of 60 R1 wild-type clones are aggregate-positive (**Figure 6—figure supplement 2A**). Starter cultures of cells containing Sup35 NM and New1 and cells containing Sup35 NM alone serve as positive (P) and negative (N) controls, respectively. In all 20 R1 wild-type clones shown, full-length (FL) ClpB is detectable, Sup35 NM fusion protein levels are comparable, and New1 fusion protein is not detectable as assessed by Western blot analysis. The α-His$_{6X}$ and α-Sup35 antibodies recognize the Sup35 NM-mCherry-His$_{6X}$ fusion protein, the α-GFP antibody recognizes the New1–mGFP fusion protein, the α-ClpB antibody recognizes the *E. coli* ClpB chaperone, and the α-RpoA antibody recognizes the α subunit of *E. coli* RNA polymerase. Cells from four aggregate-positive R1 wild-type clones visualized by fluorescence microscopy (**Figure 6—figure supplement 2C**) are indicated by asterisks. (**C**) SDS-stable Sup35 NM aggregates are not detectable in R1 Δ*clpB* clones derived from a starter culture of Δ*clpB* cells containing Sup35 NM, New1, and ectopically produced ClpB as assessed by filter retention analysis. In total, 0 of 60 R1 Δ*clpB* clones are aggregate-positive (**Figure 6—figure supplement 2B**). Starter cultures of wild-type cells containing Sup35 NM and New1 and wild-type cells containing Sup35 NM alone serve as positive (P) and negative (N) controls, respectively. In all 20 R1 Δ*clpB* clones shown, Sup35 NM fusion protein levels are comparable, and neither ClpB nor New1 fusion protein is detectable as assessed by Western blot analysis. Cells from four aggregate-negative R1 wild-type clones visualized by fluorescence microscopy (**Figure 6—figure supplement 2D**) are indicated by asterisks. (**D**) Extract prepared from cells lacking ClpB is not infectious when transformed into

*Figure 6. Continued on next page*

*Figure 6. Continued*

*S. cerevisiae* [*psi⁻*] cells. Starter cultures of wild-type cells transformed with pBR322-*SUP35 NM* and pSC101^TS-*NEW1* as well as Δ*clpB* cells transformed with pBR322-*SUP35 NM* and pSC101^TS-*NEW1-clpB* contain infectious SDS-stable Sup35 NM aggregates capable of converting [*psi⁻*] yeast cells to [*PSI⁺*]. Wild-type starter culture cells containing Sup35 NM alone lack detectable infectivity. An aggregate-positive R1 wild-type clone retains infectious Sup35 NM aggregates. In contrast, an aggregate-negative R1 Δ*clpB* clone lacks detectable infectivity, as does an aggregate-negative R1 wild-type clone. Analysis of these data by Fisher's exact test indicates that the differences in the frequency of [*PSI⁺*] transformants observed with samples containing SDS-stable Sup35 NM aggregates compared with the samples containing soluble Sup35 NM are statistically significant (p < 0.0001).

The following figure supplements are available for figure 6:

**Figure supplement 1**. Δ*clpB* cells containing New1 and ectopically produced ClpB support the formation of SDS-stable Sup35 NM aggregates.

**Figure supplement 2**. The fate of Sup35 NM in 40 wild-type R1 clones and 40 Δ*clpB* R1 clones.

mechanism underlying this reversible change in protein levels; however, we suggest that stochastic fluctuations in plasmid copy number may set the stage for such an event. We note that our experiments were performed in *recA⁻* cells, which should prevent plasmid rearrangements that might lead to a permanent loss of fusion protein coding capacity.

## Bacterial machinery capable of remodeling Sup35 NM aggregates

The question of whether or not ClpB can substitute for Hsp104 in promoting Sup35 prion propagation has been addressed in a number of studies yielding conflicting indications. On one hand, several in vitro studies have provided evidence that Hsp104 (*Shorter and Lindquist, 2004*, *2006*, *2008*; *DeSantis et al., 2012*), but not ClpB (*DeSantis et al., 2012*), can fragment amyloid aggregates in the absence of auxiliary factors. Furthermore, whereas the presence of various combinations of *S. cerevisiae* Hsp70- and Hsp40-family proteins was found to modulate Hsp104 activity on amyloid substrates (*Shorter and Lindquist, 2008*; *DeSantis et al., 2012*), ClpB appeared to remain inert even in the presence of bacterial Hsp70 (DnaK), Hsp40 (DnaJ), and nucleotide exchange factor GrpE despite exhibiting robust activity on various disordered protein aggregates in vitro (*DeSantis et al., 2012*).

On the other hand, the results of several in vivo studies suggest that ClpB, in the presence of appropriate co-chaperones, is competent to support Sup35 prion propagation in yeast (*Tipton et al., 2008*; *Reidy et al., 2012*). Based on an analysis of the in vivo activities of Hsp104/ClpB chimeras, Tipton et al. argue that prion replication in yeast requires that Hsp104 collaborate with its cognate Hsp70 chaperone system. A logical inference from their work is that the inability of ClpB to substitute for Hsp104 in supporting Sup35 prion propagation in *S. cerevisiae* is an indirect consequence of the inability of ClpB to cooperate with fungal co-chaperones. More recently, Reidy et al. provided direct support for this inference. In particular, Reidy et al. found that ClpB supported prion propagation in yeast provided that DnaK and GrpE were present. Interestingly, the activity of the bacterial disaggregase machinery in yeast was dependent on the fungal Hsp40-family Sis1 protein, consistent with prior work implicating Sis1 as a necessary component of the chaperone network required for prion propagation in yeast (*Higurashi et al., 2008*; *Tipton et al., 2008*). Our work demonstrates that no exogenous fungal accessory factors are required for prion propagation in bacteria. Taken together, these observations argue that the amyloid remodeling activity of Hsp104 is an evolutionarily conserved feature of the Hsp100-family chaperones, an inference that is strongly supported by our finding that propagation of the Sup35 NM prion in *E. coli* requires ClpB disaggregase activity.

Despite the apparent prevalence of prions in the fungal kingdom, to date, no bacterial prion has been identified. Notably, the absence of cytoplasmic mixing during conjugation would preclude the discovery of prion-like phenomena by classic genetic approaches, which facilitated the discovery of prions in yeast based on the non-Mendelian inheritance of their associated phenotypes (*Cox, 1965*; *Aigle and Lacroute, 1975*; *Wickner, 1994*). Nevertheless, recent bioinformatic analyses of prokaryotic proteomes have revealed that bacterial and archaeal genomes encode many proteins containing glutamine- and asparagine-rich prion-like domains resembling those found in most confirmed and putative *S. cerevisiae* prions (*Alberti et al., 2009*; *Espinosa Angarica et al., 2013*; Yuan et al.,

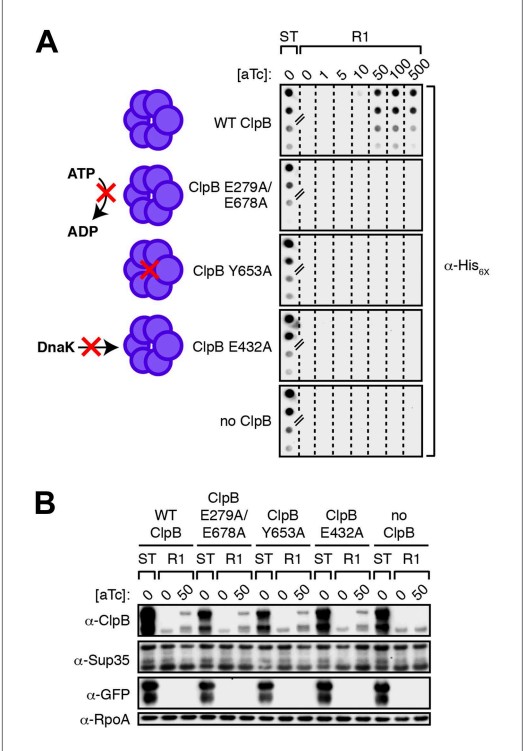

**Figure 7**. Propagation of SDS-stable Sup35 NM aggregates in *E. coli* requires ClpB disaggregase activity. (**A**) SDS-stable Sup35 NM aggregates are detected in starter cultures (ST) of Δ*clpB* cells containing pBR322-*SUP35 NM*, pSC101$^{TS}$-*NEW1-clpB*, and one of four aTc-inducible chromosomal *clpB* alleles. Wild-type (WT) ClpB is depicted as a purple hexamer. ClpB E279A/E678A is unable to hydrolyze ATP, ClpB Y653A is pore-deficient, and ClpB E432A is unable to collaborate with DnaK. Propagated Sup35 NM aggregates are detected in scraped cell suspensions as a function of increasing aTc concentration only for Round 1 (R1) clones producing wild-type ClpB. Sup35 NM aggregates are not detected at any aTc concentration in scraped cell suspensions of R1 clones producing ClpB disaggregase mutants or in R1 clones lacking ClpB. Lanes cropped from the same immunoblot are indicated by hash marks. The α-His$_{6X}$ antibody recognizes the Sup35 NM-mCherry-His$_{6X}$ fusion protein. (**B**) Wild-type and mutant ClpB levels along with Sup35 NM fusion protein levels are comparable in R1 clones grown on solid medium supplemented with 50 ng/ml aTc as assessed by Western blot analysis. The α-Sup35 antibody recognizes the Sup35 NM-mCherry-His$_{6X}$ fusion protein, the α-GFP antibody recognizes the New1-mGFP fusion protein, the α-ClpB antibody recognizes the *E. coli* ClpB chaperone, and the α-RpoA antibody recognizes the α subunit of *E. coli* RNA polymerase.

The following figure supplement is available for figure 7:

**Figure supplement 1**. All Round 1 clones producing wild-type ClpB are cured of pSC101$^{TS}$-*NEW1-clpB*.

unpublished data). Moreover, it is becoming increasingly clear that Q/N-richness at the level of primary amino acid sequence is neither a prerequisite for prion conversion (*Taneja et al., 2007*; *Suzuki et al., 2012*) nor protein amyloidogenesis (*Goldschmidt et al., 2010*). In fact, several bacteria utilize non-Q/N-rich amyloid-forming proteins to assemble extracellular appendages mediating surface attachment and biofilm formation (*Chapman et al., 2002*; *Romero et al., 2010*). These considerations—in conjunction with the work presented here—suggest that prions or prion-like proteins may exist as epigenetic reservoirs of phenotypic diversity in the bacterial domain of life.

## Materials and methods

### Strains, plasmids, and cell growth

Bacteria experiments were performed with *E. coli* strain DH5αZ1 (*Lutz and Bujard, 1997*) grown in LB (Miller) medium. To construct DH5αZ1 Δ*clpB*, a temperature-sensitive plasmid encoding the RecA protein (pSC101$^{TS}$-*recA*) was constructed and transformed into DH5αZ1 cells. A Δ*clpB*::*kan* allele from strain JW2573 (Keio collection) was transferred to DH5αZ1 cells containing pSC101$^{TS}$-*recA* via P1 transduction. Cells were subsequently cured of pSC101$^{TS}$-*recA* by overnight growth and plating in the absence of antibiotic selection at the non-permissive temperature (37°C).

To construct strains harboring chromosomal P$_{LtetO-I}$-*clpB* alleles, plasmids pAY152, pAY154, pAY155, pAY156, and pAY157 were cloned in strain AY290 and integrated onto the chromosome of strain AY295 at *attB*(HK022). Single-copy integrants were selected on LB agar supplemented with kanamycin (10 µg/ml) and verified by PCR as described (*Haldimann and Wanner, 2001*).

Yeast experiments were performed with *S. cerevisiae* strain YJW187 [*pin*$^-$][*psi*$^-$] grown in yeast extract peptone dextrose (YPD) medium. For yeast infectivity assays, cell extracts were co-transformed with pRS316 into YJW187 spheroplasts; [*PSI*$^+$] URA$^+$ transformants were identified by plating the yeast cells in top agar containing synthetic defined medium lacking uracil and adenine (SD-Ura-Ade) and supplemented with 10 mg/ml adenine hemisulfate (Sunrise Science, San Diego, CA).

Further details concerning strains and plasmids are provided in *Supplementary file 1*.

### Propagation experiments

Cells were transformed with pBR322-*SUP35 NM* and pSC101$^{TS}$, pSC101$^{TS}$-*NEW1*, or pSC101$^{TS}$-*NEW1-clpB* and grown at 30°C on LB agar

supplemented with carbenicillin (Carb, 100 µg/ml) and chloramphenicol (Cam, 12.5 µg/ml). Starter cultures were generated by growing transformants at 30°C in 6 ml of LB broth supplemented with Carb (100 µg/ml), Cam (12.5 µg/ml), and 10 µM IPTG to an $OD_{600}$ of 2.0–2.5. To cure cells of pSC101$^{TS}$-derivatives and generate Round 1 (R1) colonies, starter cultures were diluted ($10^{-5}$) in pre-warmed (37°C) LB broth supplemented with Carb (100 µg/ml) and 10 µM IPTG. Diluted cells were grown at 37°C on pre-warmed (37°C) LB agar supplemented with Carb (100 µg/ml) and 10 µM IPTG. R1–R4 colonies were, (a) patched on LB agar supplemented with Cam (12.5 µg/ml), (b) restreaked and grown at 30°C on pre-warmed (30°C) LB agar supplemented with Carb (100 µg/ml) and 10 µM IPTG, and (c) inoculated and grown at 30°C in 6 ml LB broth supplemented with Carb (100 µg/ml) and 10 µM IPTG. For analysis of ClpB disaggregase mutants, ~1000 R1 colonies were gently scraped off LB agar plates containing Carb (100 µg/ml), 10 µM IPTG, and a range of aTc concentrations (0–500 ng/ml) in 3 ml LB broth supplemented with Carb (100 µg/ml) and 10 µM IPTG.

Cell cultures and scraped cell suspensions were normalized to 8 ml of an $OD_{600}$ of 1.0 and pelleted by centrifugation. Cell pellets were resuspended in 166 ml STC Buffer (1 M sorbitol, 10 mM Tris–HCl [pH 7.5], 10 mM $CaCl_2$) supplemented with 10 U of rLysozme (Novagen, Germany) and 0.1 U of OmniCleave endonuclease (Epicentre, Wisconsin, MA), incubated at room-temperature for 30 min, and incubated on ice for an additional 30 min. Omnicleave endonuclease was omitted from samples destined for PCR analysis and yeast infectivity assays. Finally, samples were flash frozen and thawed on ice to yield unclarified lysates (used in filter retention assays). To generate partially clarified lysates (used in Western blot analysis and yeast transformation assays), unclarified lysates were subjected to two rounds of low-speed centrifugation, each at 500 RCF for 15 min at 4°C.

## Filter retention assays

25 µl of unclarified lysates was added to 375 µl of BugBuster protein extraction reagent (Novagen) supplemented with 5 U of rLysozyme and 0.1 U of Omnicleave endonuclease and gently rocked at room-temperature for 30 min. Samples were challenged with 100 µl of 10% (wt/vol) SDS (2% SDS final concentration) and gently rocked at room-temperature for an additional 30 min. For each sample, 100 µl of undiluted lysate and three twofold serial dilutions made in PBS containing 2% SDS were filtered through a 0.2-µm cellulose acetate membrane (Advantec, Japan) in a dot-blotting vacuum manifold. Samples on membranes were washed twice with 100 µl of PBS containing 2% SDS and twice with 100 µl of PBS.

## Immunoblotting

Cellulose acetate membranes (used in filter retention assays) and Hybond-C Extra nitrocellulose membranes (used in Western blot analysis) were blocked for 30 min in PBS containing 3% (wt/vol) milk. Membranes were probed with one of the following primary antibodies: anti-Sup35 (yS-20, Santa Cruz Biotechnology, Dallas, TX, 1:5000), anti-His$_{6X}$ (His-2; Roche, Indianapolis, IN, 1:10,000), anti-GFP (Roche, 1:10,000), anti-RpoA (NeoClone, Madison, WI, 1:10,000), or anti-ClpB (gift from S Wickner, 1:10,000). Membranes were washed and probed with one of the following HRP-conjugated secondary antibodies: anti-goat IgG (Santa Cruz Biotechnology, 1:10,000), anti-mouse IgG (Cell Signaling, Beverly, MA, 1:10,000), or anti-rabbit IgG (Cell Signaling, 1:10,000). Proteins were detected with ECL Plus Western blot detection reagents (GE Healthcare, Pittsburgh, PA) and a ChemiDock XRS+ imaging system (Bio-Rad, Hercules, CA).

## Yeast infectivity assays

Protein concentrations of partially clarified *E. coli* cell extracts were determined by the bicinchoninic acid (BCA) assay (ThermoFisher, Waltham, MA) and normalized to ~1 mg/ml. Protein transformations were performed as previously described (*Tanaka and Weissman, 2006*; *Garrity et al., 2010*). Each and every putative [*PSI*$^+$] transformant was (a) restreaked on 1/4 YPD agar to assess the [*PSI*$^+$] phenotype, (b) restreaked on YPD containing 3 mM GuHCl to cure cells of [*PSI*$^+$], and (c) restreaked on 1/4 YPD to assess the [*psi*$^-$] phenotype. Only those transformants exhibiting curability were scored as [*PSI*$^+$].

## Fluorescence microscopy

Cells were spotted onto 1% (wt/vol) agarose pads consisting of Seakem LE Agarose (Lonza, Walkersville, MD) in PBS and visualized with an UplanFL N 100x/1.30 phase contrast objective

mounted on an Olympus BX61 microscope. Images were captured with a CoolSnapHQ camera (Photometrics, Tucson, AZ) and the Metamorph software package (Molecular Devices, Sunnyvale, CA). All fluorescence images were obtained from 10 ms exposures.

## Acknowledgements
We thank Simon Dove (Harvard Medical School, Boston, MA) for comments on the manuscript, Sue Wickner (National Cancer Institute, Bethesda, MD) for anti-ClpB serum, and Renate Hellmiss for preparing *Figure 2*.

## Additional information

### Funding

| Funder | Grant reference number | Author |
| --- | --- | --- |
| National Institutes of Health | Pioneer award, OD003806 | Ann Hochschild |
| National Science Foundation | Graduate research fellowship | Andy H Yuan |

The funders had no role in study design, data collection and interpretation, or the decision to submit the work for publication.

### Author contributions
AHY, Conception and design, Acquisition of data, Analysis and interpretation of data, Drafting or revising the article; SJG, EN, Conception and design, Contributed unpublished essential data or reagents; AH, Conception and design, Analysis and interpretation of data, Drafting or revising the article

## Additional files

### Supplementary file
• Supplementary file 1. Strains and Plasmids.

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
