## [Decision Letter]

[Editors’ note: this article was originally rejected after discussions between the reviewers, but the authors were invited to resubmit after an appeal against the decision.]

Thank you for choosing to send your work entitled “Prion propagation can occur in a prokaryote and requires the ClpB chaperone” for consideration at *eLife*. Your full submission has been evaluated by K (Vijay) VijayRaghavan (Senior editor) and 3 peer reviewers, one of whom, Peter Greenberg, is a member of our Board of Reviewing Editors, and the decision was reached after discussions between the reviewers.

The reviewers all thought this was an elegantly executed study but in the end that it did not constitute enough of an advance to merit publication in *eLife*. The specific reviews are self-explanatory. We are sorry we do not have better news at this time.

Reviewer #1:

The authors build on a previous study they published in which they showed that the yeast prion domain (Sup35NM) can take up an infectious prion form in a bacterial (*E. coli*) cell. In this important new paper they now go on to show that this prion form can be propagated over a large number of cell divisions in *E. coli* and that this propagation requires the Hsp104 bacterial ortholog of Hsp104, i.e. ClpB, to do so. They provide a very detailed description of their experimental strategy and, although the ability of *E. coli* to propagate the prion form of Sup35NM is certainly less efficient than one sees in yeast, nevertheless their data are convincing that propagation does indeed require ClpB. What they do not formally demonstrate is that it is the chaperone activity of ClpB that is important and this could be easily addressed by repeating their experiments with an appropriate ClpB mutant.

The chaperone activity of ClpB depends on its oligomerisation and its ATPase activity and numerous chaperone-defective mutants have been described. Such an experiment should be straightforward and well within the grasps of the authors. Given the important implications of their findings, i.e. that *E. coli* has the chaperone machinery to propagate a (heterologous) prion; I think it necessary that they demonstrate that it is indeed the chaperone activity that is essential. Of course it remains to be established if *E. coli* (or any other bacterial spp) has endogenous prion-forming proteins but the findings reported here give one confidence that they are there, we just have to find a way of detecting them. By showing the dependency on ClpB, the authors open up one strategy to do so.

Reviewer #2:

The manuscript submitted by Hochschild and colleagues addresses the ability of *E. coli* to support propagation of the [PSI+] prion of yeast through the self-assembly and inheritance of aggregates of the prion-determining domain (NM) of the Sup35 protein. The significance of this work to the field and its ability to advance the current state of knowledge are somewhat limited:

1) Previous work from Hochschild and colleagues (Garrity et al. PNAS 2010) demonstrated that NM can form SDS-resistant aggregates that, when introduced into yeast cytosol, induce the appearance of [PSI+]. In the current work, the authors reveal that this state is maintained in *E. coli* through multiple generations and that its propagation, unlike its initiation, is independent of another yeast prion protein New1. These principles have been previously established in yeast, and it’s unclear how their extension into *E. coli* advances our understanding of the process or its evolutionary conservation.

2) The authors demonstrate that ClpB, the bacterial homologue of Hsp104, is required for propagation of NM aggregates in *E. coli*. Previous work from the Masison lab has demonstrated that, when expressed in yeast, ClpB and its Hsp70 co-chaperone DnaK are sufficient to support [PSI+], so the fact that ClpB can do so within its native environment is not surprising. Moreover, this result simply reports a genetic interaction without mechanistic insight. Is ClpB performing the same function in *E. coli* as Hsp104 does in yeast?

Together, these studies only represent an incremental advance over previous work in the field.

Reviewer #3:

Yuan et al present evidence for the establishment of yeast prion propagation in the prokaryote *E. coli*. This is dependent on the presence of both the yeast PIN factor New1 and the bacterial Hsp104 homologue ClpB. A PIN factor is required for conversion of Sup35 from psi- to PSI+ (the prion form) and Hsp104 has long been known to be required for the propagation of nearly all yeast prions known. Thus the key finding of this carefully done paper is that ClpB the E coli homologue of yeast Hsp104 is also analogous (having the same function). Since most homologues are analogues and many yeast proteins can function in *E. coli*, at one level this result is not too surprising. It does however suggest that fundamentally there are no huge differences between the cytosol of yeast and *E. coli* in regards to prion propagation apart from the presumable absence of PIN factors in *E. coli*. This work sets the stage for further genetic analysis of yeast prion propagation in *E. coli*.

Since both of these systems are genetically well characterized some overall conclusions about the fundamentals of in vivo prion propagation may well be able to be reached in future work, though such work is prone to false negative results as there can be many trivial reasons for lack of functional complementation when working in such evolutionarily distant systems. The authors make a distinction between prion establishment and prion propagation, but these are really more tightly linked as one cannot establish a prion without propagating it. They do come out on the side of ClpB and New1 both being required for prion propagation but indeed since both were simultaneously depleted in the experiments shown in Figure 6, it seems more likely that they are both required for both the establishment and propagation phases. Even given the fuzziness of the distinction is not clear if ClpB is required for the propagation alone or if it is just required for prion establishment. Does expression of NM make *E. coli* sick? If it does then that raises the possibility that strains that propagate prions are able to sequester it as a less toxic form. This in combination with the observation that the vast majority of ClpB+ colonies fail to propagate NM raises the possibility that *E. coli* host mutations are also necessary for prion propagation and that *E. coli* is coming up with various other ways to get around this sickness. The various forms of aggregates observed raises the issue of which are prion-like and which are non-propagating. The real definition of prion I guess is ability to propagate, more than its SDS insolubility. This distinction should be made clearer throughout the manuscript.

---

## [Author Response]

Reviewer #1:

*[…] They provide a very detailed description of their experimental strategy and, although the ability of E. coli to propagate the prion form of Sup35NM is certainly less efficient than one sees in yeast, nevertheless their data are convincing that propagation does indeed require ClpB. What they do not formally demonstrate is that it is the chaperone activity of ClpB that is important and this could be easily addressed by repeating their experiments with an appropriate ClpB mutant*.

*The chaperone activity of ClpB depends on its oligomerisation and its ATPase activity and numerous chaperone-defective mutants have been described. Such an experiment should be straightforward and well within the grasps of the authors. Given the important implications of their findings, i.e. that E. coli has the chaperone machinery to propagate a (heterologous) prion; I think it necessary that they demonstrate that it is indeed the chaperone activity that is essential. Of course it remains to be established if E. coli (or any other bacterial spp) has endogenous prion-forming proteins but the findings reported here give one confidence that they are there, we just have to find a way of detecting them. By showing the dependency on ClpB, the authors open up one strategy to do so*.

We fully agree with the reviewer that our study would be enhanced by demonstrating that it is in fact the disaggregase activity of ClpB that is required for the propagation of the Sup35-NM prion in *E. coli*. The reviewer suggests that it would be relatively straightforward to address this point by using appropriate ClpB mutants. Indeed, we have already initiated such experiments and aim to test two well characterized ClpB mutants that are (i) specifically deficient in ATPase activity and (ii) specifically defective in substrate translocation through the ClpB pore. We hypothesize that neither of these mutants will support propagation of the Sup35-NM prion.

Reviewer #2:

*The manuscript submitted by Hochschild and colleagues addresses the ability of E. coli to support propagation of the [PSI+] prion of yeast through the self-assembly and inheritance of aggregates of the prion-determining domain (NM) of the Sup35 protein. The significance of this work to the field and its ability to advance the current state of knowledge are somewhat limited. […] Together, these studies only represent an incremental advance over previous work in the field*.

In response to reviewer #2, we wish to clarify why we believe our findings are broadly significant. Specifically, our findings establish that prion biology can operate in the bacterial domain of life and thus they provide the first demonstration of protein-based inheritance in a non-eukaryote. Additionally, as pointed out by reviewer #1, our study lays the foundation for a search for endogenous prion-like mechanisms in bacteria (reviewer #1 says, “the findings reported here give one confidence that they [endogenous bacterial prion proteins] are there, we just have to find a way of detecting them. By showing the dependency on ClpB, the authors open up one strategy to do so”). The second reviewer’s assessment that our study represents only “an incremental advance of previous work in the field” appears to be in reference to the field of fungal prion biology. We agree with the reviewer that our findings do not significantly deepen the mechanistic understanding of prion propagation in the endogenous fungal system. Rather we believe that the significance of our work is that we have uncovered a foundation for prion biology in another domain of life.

The reviewer also suggests that our findings with regard to ClpB are “not surprising”. However, given a prominently published study from the Shorter lab, which concludes, based on experiments performed *in vitro*, that ClpB is incapable of remodeling prion-like amyloid aggregates, we believe our findings will be surprising to at least some investigators in the field. (In demonstrating that ClpB is capable of remodeling prion-like aggregates, our findings complement those from the Masison lab, which we discuss; we note, however, that Masison and colleagues found that the ability of ClpB to support [PSI+] in yeast was dependent on the fungal Hsp40-family Sis1 protein, whereas our work shows that no fungal accessory factors are required in the endogenous bacterial setting.) Specifically, our results and those of the Masison lab challenge a prevalent view in the yeast prion field – namely that the ability to remodel prion-like aggregates is an Hsp104 innovation specific to fungal systems. Instead, our results raise the possibility that this ability predates the split between bacteria and yeast and has been conserved in their respective chaperone systems.

Finally, the reviewer asks if ClpB is performing the same function in *E. coli* as Hsp104 does in yeast. As discussed above (response to reviewer #1), we agree that this is an important question and are currently performing experiments to address this point.

Reviewer #3:

*Yuan et al present evidence for the establishment of yeast prion propagation in the prokaryote E. coli. This is dependent on the presence of both the yeast PIN factor New1 and the bacterial Hsp104 homologue ClpB. A PIN factor is required for conversion of Sup35 from psi- to PSI+ (the prion form) and Hsp104 has long been known to be required for the propagation of nearly all yeast prions known. Thus the key finding of this carefully done paper is that ClpB the E coli homologue of yeast Hsp104 is also analogous (having the same function). Since most homologues are analogues and many yeast proteins can function in E. coli, at one level this result is not too surprising. It does however suggest that fundamentally there are no huge differences between the cytosol of yeast and E. coli in regards to prion propagation apart from the presumable absence of PIN factors in E. coli. This work sets the stage for further genetic analysis of yeast prion propagation in E. coli*.

*Since both of these systems are genetically well characterized some overall conclusions about the fundamentals of in vivo prion propagation may well be able to be reached in future work, though such work is prone to false negative results as there can be many trivial reasons for lack of functional complementation when working in such evolutionarily distant systems. The authors make a distinction between prion establishment and prion propagation, but these are really more tightly linked as one cannot establish a prion without propagating it. They do come out on the side of ClpB and New1 both being required for prion propagation but indeed since both were simultaneously depleted in the experiments shown in*
Figure 6*, it seems more likely that they are both required for both the establishment and propagation phases. Even given the fuzziness of the distinction is not clear if ClpB is required for the propagation alone or if it is just required for prion establishment. Does expression of NM make E. coli sick? If it does then that raises the possibility that strains that propagate prions are able to sequester it as a less toxic form. This in combination with the observation that the vast majority of ClpB+ colonies fail to propagate NM raises the possibility that E. coli host mutations are also necessary for prion propagation and that E. coli is coming up with various other ways to get around this sickness. The various forms of aggregates observed raises the issue of which are prion-like and which are non-propagating. The real definition of prion I guess is ability to propagate, more than its SDS insolubility. This distinction should be made clearer throughout the manuscript*.

The reviewer suggests that the distinction between prion establishment and prion propagation is a fuzzy one. We suspect that we did not make the distinction clear enough in our manuscript. We believe (i) that there is a clear conceptual distinction between the *de novo* formation phase and the propagation phase and (ii) that our *E. coli*-based system provides us with a rigorous experimental strategy to distinguish between these two phases. Conceptually, the *de novo* formation of prion-like aggregates from soluble protein within a given cell (which can be facilitated by protein overproduction) represents “establishment”, whereas the maintenance of these aggregates in progeny cells under conditions that do not permit *de novo* formation represents “propagation”. Accordingly, the strict [PIN+] dependence of Sup35-NM aggregate formation in our bacterial system allows for a clean separation between a *de novo* formation phase (which occurs in our “starter” cultures) and a propagation phase (which occurs after plating). Specifically, the detection of prion-like aggregates in cells that have arisen multiple generations after the loss of the vector encoding New1 (the source of [PIN+] in our system) is evidence for propagation (i.e. protein-based inheritance).

Based on these considerations (which we believe we can articulate more clearly in a revised text), we disagree with the reviewer’s comment that it is “not clear if ClpB is required for propagation alone or if it is just required for prion establishment”. If ClpB were required just for prion establishment, then depleting the cells of ClpB *after* the establishment phase would not have resulted in Round-1 progeny clones that uniformly lack prion-like aggregates (see Figure 6).

We agree with the reviewer’s last substantive point: “The real definition of prion I guess is ability to propagate, more than its SDS insolubility. This distinction should be made clearer throughout the manuscript”. We will carefully revise the text to avoid confusion on this critical point.